# Streaming Long Video Understanding with Large Language Models

**Rui Qian**[1*] **Xiaoyi Dong**[1,2†] **Pan Zhang**[2] **Yuhang Zang**[2] **Shuangrui Ding**[1]
**Dahua Lin**[1,2,3] **Jiaqi Wang**[2†]
[1] The Chinese University of Hong Kong
[2] Shanghai AI Laboratory
[3] HKGAI under InnoHK

## Abstract

This paper presents **VideoStreaming**, an advanced vision-language large model (VLLM) for video understanding, that capably understands arbitrary-length video with a constant number of video tokens streamingly encoded and adaptively selected. The challenge of video understanding in the vision language area mainly lies in the significant computational burden caused by the great number of tokens extracted from long videos. Previous works rely on sparse sampling or frame compression to reduce tokens. However, such approaches either disregard temporal information in a long time span or sacrifice spatial details, resulting in flawed compression. To address these limitations, our VideoStreaming has two core designs: Memory-Propagated Streaming Encoding and Adaptive Memory Selection. **The Memory-Propagated Streaming Encoding architecture** segments long videos into short clips and sequentially encodes each clip with a propagated memory. In each iteration, we utilize the encoded results of the preceding clip as historical memory, which is integrated with the current clip to distill a condensed representation that encapsulates the video content up to the current timestamp. This method not only incorporates long-term temporal dynamics into the streaming encoding process but also yields a fixed-length memory as a global representation for arbitrarily long videos. After the encoding process, the **Adaptive Memory Selection strategy** selects a constant number of question-related memories from all the historical memories, and feeds them into the LLM to generate informative responses. The question-related selection reduces redundancy within the memories, enabling efficient and precise video understanding. Meanwhile, the disentangled video extraction and reasoning design allows the LLM to answer different questions about a video by directly selecting corresponding memories, without the need to encode the whole video for each question. Through extensive experiments, our model achieves superior performance and higher efficiency on long video benchmarks, showcasing precise temporal comprehension for detailed question answering.

## 1 Introduction

The evolution of Large Language Models (LLMs) has significantly advanced artificial intelligence, encompassing text generation and reasoning in complex language environments [9, 74, 14, 61, 16, 69, 2, 70]. Later, the community extends LLMs to multi-modal domains, demonstrating promising results in captioning and question-answering tasks that integrate diverse visual signals [46, 40, 15, 59]. Yet, within the domain of video understanding, long video sequences pose a formidable challenge.

---

[*]Work done during an internship in Shanghai AI Laboratory.
[†]Corresponding Author.

38th Conference on Neural Information Processing Systems (NeurIPS 2024).

Incorporating such long visual contents into LLMs requires a substantial number of tokens, which not only amplifies computational demands but also risks early contextual information loss [49].

Among the recent works on general video understanding with LLMs [48, 42, 44, 88, 50, 43, 66, 62], a prevalent strategy is using sparse temporal sampling [44, 86] or spatio-temporal pooling [50, 48] to reduce tokens. Unfortunately, this paradigm explicitly loses substantial information in the long time span. To address this limitation, [43, 42, 88] develop frame-wise compression, with LLaMA-VID [43] as a typical example. It compresses each frame into only two tokens but overlooks the inter-frame temporal dynamics which are vital in compressing temporal redundancy within videos. Besides, its question-dependent compression pipeline limits the ability to produce a general representation that can handle diverse instructions. Another line of works employ memory banks [77, 7] to store history information [66, 22]. Whereas, these methods rely on explicit timestamps to recall the historical details, limiting the ability to generate comprehensive responses without specific time indicators.

In this work, we propose VideoStreaming, a novel Memory-Propagated Streaming Encoding architecture with Adaptive Memory Selection to sequentially encode a long video into condensed memories and generate responses referring to relevant timestamps. The core idea behind the memory-propagated streaming encoding is to preserve representative spatial cues and temporal dynamics while reducing temporal redundancy in videos. To achieve this goal, we segment the long video into multiple short clips and sequentially encode each clip. When encoding each clip, we first refer to the encoded results of its preceding clip as historical memory, then concatenate it with the current clip features and feed them into a small decoder-only language model [20]. Due to its autoregressive nature, the information of the sequence naturally accumulates to the last few tokens [38, 34]. Consequently, we take these last few tokens as an updated memory that encapsulates the video information up to the current timestamp. Through this streaming encoding, we explicitly take long-term temporal relations into consideration and maintain a fixed-length memory to represent an arbitrarily long video.

However, this fixed-length memory inevitably loses detailed information, especially in early contexts. To address this problem, we store the historical memories of all clips and select a constant number of subsets that are closely related to the question. To accomplish this, when streaming encoding each clip, we additionally append a summary token at the end of the sequence as a clip indicator that summarizes the clip contents within one token. Then, given a specific question, we concatenate the condensed memory from the final iteration with the question and pass it through the same small language model used in streaming encoding. We take the final token as the question indicator and calculate its similarity with all historical clip indicators, the clip indicator with higher similarity means its corresponding memory is more related to the question. Finally, we feed the adaptively selected memories into the LLM for detailed question answering.

In practice, we realize our VideoStreaming with a carefully designed two-stage progressive training process and long-video data construction strategy. In the first stage, we empower a small language model with the single-clip encoding capability by a specialized prefix task. In the second stage, it serves as the streaming encoder and we jointly train it with the LLM for long video understanding. Due to the lack of long video QA data, we manually constructed a set of long video QA pairs in two ways. On the one hand, we concatenate short videos from existing datasets [76, 81] into longer ones, where the original questions correspond to different segments. On the other hand, we curate a subset of Panda-70M [11] which includes captions for segmented clips as well as the original long videos, and use this to create multi-round long video QA pairs with explicit timestamps. These long video QA data not only optimize the responses from the LLM but also guide accurate memory selection.

In summary, our contributions are as follows: (1) We analyzed the challenge of long video understanding in the vision language area, and pointed out that the problem of current methods lies in the inefficient video encoding. (2) In response to the challenges, we propose two efficient designs: Memory-Propagated Streaming Encoding and Adaptive Memory Selection, which result in our advanced video understanding model VideoStreaming. (3) The extensive experiments demonstrate that our model achieves precise temporal grounding with respect to specific questions, attains superior performance, and exhibits higher inference efficiency on long video benchmarks.

## 2   Related Work

**Large Language Models** (LLMs) have revolutionized natural language processing. Early works establish encoder-decoder models with masked language modeling [16, 61], while later decoder-only

models like GPT [60] showcase remarkable performance and scalability. Recent groundbreaking works, such as PaLM [14], LLaMA [70] and GPT-4 [56], have pushed the boundaries by developing significantly larger models with billions of parameters. To harness the full potential of LLMs, a series of works [55, 57, 13] adopt supervised instruction tuning [74] to guide models towards generating more natural and contextually relevant responses. Inspired by the powerful reasoning capacities of LLMs, we explore using LLMs for challenging long video understanding.

**Vision Language Models** like CLIP [59] employ contrastive learning on image-text pairs to formulate a unified embedding space [59, 28, 40]. Later, [46, 39, 56, 90, 3, 4, 85] integrate image features into LLMs and achieve promising visual reasoning. To process more complex video data, [50, 48, 44, 25, 86] use sparse sampling or simple temporal pooling to obtain compact video tokens for LLMs. [42, 88] employ Q-Former [39] to project frame-wise features into the textual space. To handle longer videos, [31, 75] utilize token merging [8] to reduce redundancy and alleviate computational burden. LLaMA-VID [43] proposes an instruction-aware compression strategy to represent each frame with only two tokens, but it overlooks the temporal relations in the compression step. [66, 22] develop memory banks to accumulate information in long videos and excel in global video comprehension. However, these methods struggle with moment-specific questions without explicit time indicators. To address these limitations, we propose a memory-propagated streaming encoding architecture with adaptive memory selection, which effectively reduces temporal redundancy and accurately selects relevant information for detailed question answering.

**Long Video Understanding** is a challenging task in computer vision. The most prevalent strategy is to maintain a memory bank to store history information in long videos [77, 7, 78, 54, 12, 71, 21]. Typically, MemDPC [21] formulates a memory bank shared across the whole dataset as an external knowledge reference for future prediction, but it is not explicitly designed for long-term understanding. To facilitate long video content analysis and ensure computation efficiency, MC-ViT [7], MovieChat [66] and StreamingCaption [89] rely on handcrafted rules like clustering to compress the history into a finite-length memory and iteratively update it in video streams. A potential drawback is that the memory cannot be optimized through the extensive video caption data. It is potentially more promising to learn comprehensive memory consolidation in a data-driven manner. More recently, [32, 27, 87] use language as a bridge for long-term video understanding. They first divide a long video into short clips, generate textual descriptions for each clip, and then employ an LLM to aggregate the short captions for long video analysis. However, this architecture cannot be trained end-to-end, and the long video understanding quality depends on the short clip captions. In contrast, we employ a trainable small language model to iteratively encode short clips into compact memories, which can be jointly optimized with the subsequent LLM for long video understanding.

## 3 VideoStreaming

In this section, we introduce VideoStreaming, a streaming long video understanding framework with LLM. As illustrated in Fig. 1a, given a long video input, VideoStreaming segments it into multiple short clips and iteratively encodes each clip into compact historical memory. To enhance the reasoning ability to specific questions, we design an adaptive memory selection strategy to select a subset of relevant memories and feed them into an LLM to produce detailed responses.

### 3.1 Single Clip Encoding

To effectively distill the information within a sequence into a compact set of tokens, we take inspiration from recent advanced decoder-only language models [2, 70, 13, 5, 29] and employ a comparatively small language model, Phi-2 [20], for efficient encoding. Due to the causal attention and autoregressive nature, it is intuitive to utilize a language model to aggregate the sequence information onto the last few tokens [38, 34], which naturally serve as a compact representation that provides a high-level summary of the input sequence.

Mathematically, given a $T$-frame video clip, we first use a pre-trained CLIP ViT-L [59] to extract frame-wise features and concatenate every four spatially adjacent visual tokens along channel dimension to reduce the number of tokens by 75%. The resulting clip features are denoted as $\boldsymbol{F} \in \mathbb{R}^{TN \times C}$, where $N$ denotes the per-frame spatial token number, and $C$ is the channel dimension. To produce the condensed representations, we initialize a set of summarization tokens $\boldsymbol{S} \in \mathbb{R}^{TP \times C}$ by adaptively pooling each frame into $P$ tokens, where $P \ll N$. Intuitively, $\boldsymbol{S}$ can be regarded as a

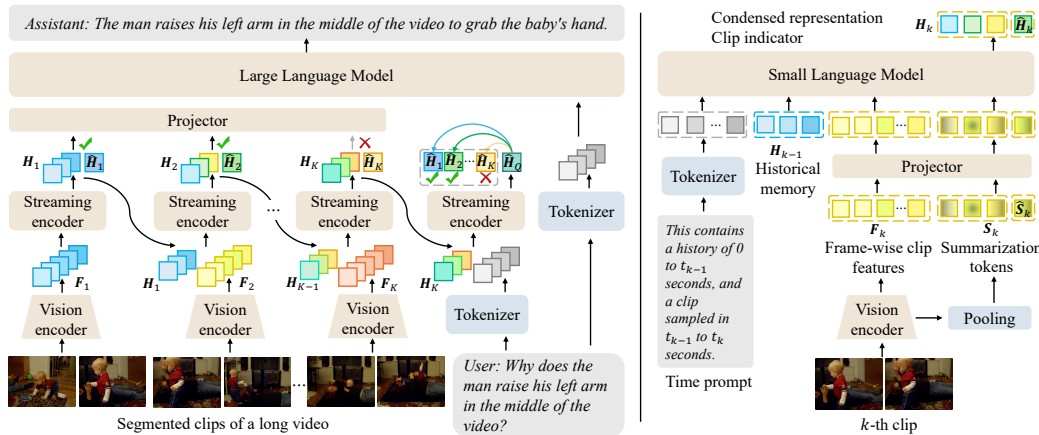

(a) An overview of streaming video understanding framework.  (b) Details of the streaming encoder.

Figure 1: Fig. 1a shows an overview of VideoStreaming, where we segment a long video into short clips and iteratively encode each clip into compact memories. Then, according to specific questions, we select a constant number of subsets of relevant memories as input to an LLM to produce responses. The ✓ and ✗ respectively denote selected and unselected memories. Fig. 1b illustrates the detailed process of each streaming encoding iteration. We encode current clip features with reference to specific timestamps and historical memory from the preceding clip into a condensed representation.

coarse encapsulation of the given clip, making it well-suited to serve as the summarization tokens for consolidating the clip information. To this end, we concatenate $\boldsymbol{F}$ with $\boldsymbol{S}$ and feed them into the encoder $g(\cdot)$, which consists of an MLP projector and a language model Phi-2. We utilize the output of the last $T \times P$ tokens as the condensed representation of the given clip:

$$\boldsymbol{H} = g([\boldsymbol{F} \circ \boldsymbol{S}]) \in \mathbb{R}^{TP \times D}, \tag{1}$$

where $\circ$ denotes concatenation operation, $D$ is the channel dimension of Phi-2.

To reinforce the visual consolidation ability, we design a prefix task to train the encoder on visual captioning and question-answering tasks. In particular, to guarantee that the clip information is distilled into the summarization tokens, we enforce the language model to generate the response only with reference to these few tokens. To achieve this goal, a straightforward way is to modify the attention mask in each Transformer decoder layer. As depicted in Fig. 2, we take a sequence covering $TN$ clip feature tokens, $TP$ summarization tokens, and $TT$ text response tokens as an example. Based on the standard causal attribute, the binary attention mask $\mathbf{M}$ is modified as shown in Figure 3:

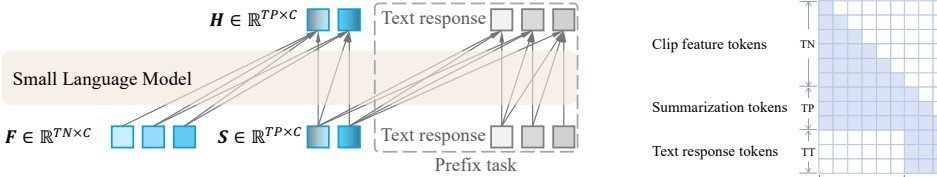

Figure 2: Illustration of the prefix task format.  Figure 3: Modified attention mask $\mathbf{M}$.

with the modified attention mask, the $TT$ text tokens can only get video-related information from the $TP$ summarization tokens to predict the next token. This encourages the summarization tokens to extract more video information from previous $TN$ video clip tokens, *ie.* learns better video encoding.

## 3.2 Memory-Propagated Streaming Long Video Encoding

Till this point, we have obtained an encoder capable of distilling short video clips into condensed representations. The next step is to comprehensively consider the long-term temporal relations within the complete videos, leveraging the historical information from previous clips to facilitate the encoding of subsequent segments as depicted in Fig. 1b.

To accomplish this objective, we divide a long video into $K$ clips, each containing $T$ frames, and propose a memory-propagated streaming encoding mechanism to iteratively encode each clip in

sequence. In each iteration, we employ the encoded results from the last iteration as historical memory and integrate them with current clip features to produce an updated memory for subsequent encoding. Specifically, given the $k$-th clip, we denote the current clip features as $\boldsymbol{F}_k \in \mathbb{R}^{TN \times C}$, the summarization tokens as $\boldsymbol{S}_k \in \mathbb{R}^{TP \times C}$, and an additional global token as $\hat{\boldsymbol{S}}_k \in \mathbb{R}^{1 \times C}$. This global token, initialized by global average pooling on the clip features $\boldsymbol{F}_k$, is expected to summarize the entire clip contents and serve as a clip indicator for memory selection in the next subsection. To enrich the temporal contexts, we refer to the encoded representations from the previous clip $\boldsymbol{H}_{k-1} \in \mathbb{R}^{TP \times D}$ to provide historical information. Then we jointly feed them into the streaming encoder to produce the condensed representation $\boldsymbol{H}_k \in \mathbb{R}^{TP \times D}$ and the clip indicator $\hat{\boldsymbol{H}}_k \in \mathbb{R}^{1 \times D}$ of the $k$-th clip:

$$\boldsymbol{H}_k, \hat{\boldsymbol{H}}_k = g([\boldsymbol{H}_{k-1} \circ \boldsymbol{F}_k \circ \boldsymbol{S}_k \circ \hat{\boldsymbol{S}}_k]). \tag{2}$$

Note that for the first clip encoding, the historical memory is not used. Through this streaming encoding process, $\boldsymbol{H}_k$ not only encompasses the current clip information but encapsulates the overall video content up to the $k$-th clip. To this end, we manage to maintain a fixed length of memory to represent arbitrarily long videos.

**Discussion.** In this architecture, we use a language model for video encoding, which has the unique advantage that we can flexibly provide the encoder with diverse prompts to guide the encoding process. Hence, the summarization tokens capture not only the core content but also additional contextual information. Typically, the explicit timestamp is an important cue in videos [62]. As shown in Fig. 1b, we incorporate a text prompt indicating the specific timestamps of each clip and historical memory to enhance temporal awareness. Besides, this prompt-based approach also allows the user to tailor the condensed output to better suit the needs of downstream tasks, going beyond a purely extractive summarization.

Another noteworthy point is that in the language model, the feature space of the final decoder layer is designed for the next token prediction, which may not perfectly align with the objective of producing condensed video representations. Considering that we modify the attention masks in each decoder layer to encourage information consolidation, this allows us to leverage the intermediate outputs from partial attention layers as the encoded results. Similar to the techniques in vision domain [82, 46, 17], this strategy potentially enables the model to capture a richer set of semantic and contextual features as the condensed representations, bridging the gap between the language model's original training objective and the requirements for video encoding.

### 3.3 Adaptive Memory Selection

Through the streaming video encoding, it is feasible to use the encoded results from the final iteration, i.e., $\boldsymbol{H}_K$, as a compact global memory that concludes the entire video. However, this fixed-length memory inevitably loses details, especially the information from early segments. Hence, this global memory alone is insufficient for comprehensive long video understanding.

To address this limitation, we make use of the encoded results of all historical clips of the input video, i.e., $\mathcal{H} = \{\boldsymbol{H}_1, \boldsymbol{H}_2, ..., \boldsymbol{H}_K\}$. Given a specific question or instruction, we first generate an adaptive indicator that summarizes relevant video content for that particular instruction. We accomplish this by reusing the language model in the streaming encoder, where we concatenate the global memory from the final iteration, $\boldsymbol{H}_K$, and the instruction texts, then pass the sequence into the model. We employ the output of the final token as the instruction indicator, denoted as $\hat{\boldsymbol{H}}_Q \in \mathbb{R}^{1 \times D}$. Thereafter, we calculate the cosine similarity between this instruction indicator and all historical clip indicators $\{\hat{\boldsymbol{H}}_1, \hat{\boldsymbol{H}}_2, ..., \hat{\boldsymbol{H}}_K\} \in \mathbb{R}^{K \times D}$ and obtain the similarity distribution $\boldsymbol{s} \in \mathbb{R}^K$. To achieve a differentiable discrete selection, we adopt Gumbel-Topk technique [36] to produce a binary index $\mathbf{I}$ that activates a subset of $V$ out of $K$ positions with the highest similarities:

$$\mathbf{I} = \text{Gumbel-Topk}(\boldsymbol{s}, V) \in \{0, 1\}^K. \tag{3}$$

Based on $\mathbf{I}$, we select the corresponding encoded results from $\mathcal{H}$ to formulate a subset of memories that are related to the instruction:

$$\hat{\mathcal{H}} = \{\mathbf{I}_k \cdot \boldsymbol{H}_k | \mathbf{I}_k = 1\}, \tag{4}$$

where $\mathbf{I}_k$ denotes the selected indexes. We concatenate the selected memories $\hat{\mathcal{H}}$ in temporal order, resulting in a sequence consisting of $V \times T \times P$ tokens. Then, we feed the sequence with instruction texts into an LLM for comprehensive reasoning.

Table 1: The statistics of the average video duration time of each evaluation dataset.

| Dataset | Duration |
| --- | --- |
| Next-QA [79] | 42.23 sec |
| Next-GQA [80] | 39.60 sec |
| VideoChatGPT [50] | 1.81 min |
| EgoSchema [51] | 3.00 min |
| MovieChat-1K [66] | 7.66 min |
| MovieNet-QA [68] | 108.26 min |

Table 2: Results on VideoChatGPT benchmark [50].

| Method | Params | CI | DO | CU | TU | CO |
| --- | --- | --- | --- | --- | --- | --- |
| Video-LLaMA [88] | 7B | 1.96 | 2.18 | 2.16 | 1.82 | 1.79 |
| VideoChat [42] | 7B | 2.23 | 2.50 | 2.53 | 1.94 | 2.24 |
| VideoChatGPT [50] | 7B | 2.40 | 2.52 | 2.62 | 1.98 | 2.37 |
| MovieChat [66] | 7B | 2.76 | 2.93 | 3.01 | 2.24 | 2.67 |
| LongVLM [75] | 7B | 2.76 | 2.86 | 3.34 | 2.39 | 3.11 |
| LLaMA-VID [43] | 13B | 3.07 | 3.05 | 3.60 | 2.58 | 2.63 |
| PLLaVA [83] | 13B | 3.27 | 2.99 | 3.66 | 2.47 | 3.09 |
| Ours | 7B+1.3B | **3.33** | **3.27** | **3.73** | **2.74** | **3.15** |

Our adaptive memory selection allows the model to dynamically access historical memories relevant to specific instructions, which mitigates the information loss inherent in the streaming encoding process. By drawing upon fine-grained details across the full video duration, the LLM can provide detailed and informative responses, while preserving high computational efficiency.

### 3.4 Short-to-Long Training

To train VideoStreaming, we design a progressive two-stage paradigm. First, we train single clip encoding on image and short video understanding tasks. Next, we train memory-propagated streaming encoding and adaptive memory selection as well as the LLM for long video understanding.

**Single Clip Training.** In this stage, both image- and video-text pairs are used to train the encoder to handle general visual signals. Following [44, 50, 88, 43], we employ 790K image and short video caption data [64, 6] to train the MLP projector for modality alignment. After that, we employ 763K image and video instruction data from [46, 50, 45] to finetune the small language model. For video input, we uniformly sample $T = 16$ frames with spatial resolution $224 \times 224$ and use a frozen CLIP ViT-L/14 [59] to extract frame-wise features. After adjacent token merging, we obtain $16 \times 64 = 1024$ tokens as the clip feature representation. Then, the encoder, a two-layer MLP and a small language model Phi-2 2.7B [20], distills each frame into $P = 4$ tokens, resulting in $16 \times 4 = 64$ tokens as the condensed representation with a compression ratio of $16 : 1$. For image-text pairs, we regard the images as single-frame clips and encode each into $4$ tokens. We use standard next token prediction to consolidate visual contents into compact summarization tokens as illustrated in Fig. 2.

**Streaming Long Video Training.** In the second stage, we use long video QA pairs to finetune the whole architecture, including ViT, the streaming encoder, and the LLM, as shown in Fig. 1a. The long video QA data encompasses three parts. (1) We adopt 25K movie QA pairs from [43, 24, 66]. (2) We curate a subset from Panda-70M [11], which provides the original long videos and the captions of segmented clips. Based on this subset, we create 300K multi-round long video QA pairs with explicit timestamps. (3) We synthesize 20K long videos by concatenating short videos from existing QA datasets [81, 76], and the original QA pairs correspond to different segments in the synthesized long videos. For each video, we extract 16-frame clips at 1 FPS, and the number of clips varies with the video duration. In streaming encoding, we employ the intermediate outputs from the first 16 layers of Phi-2 as the condensed memories. Finally, we select $V = 4$ most relevant timestamps and feed the selected memories of $V \times T \times P = 256$ tokens into the LLM, Vicuna-7B [13], for long video reasoning. Since our curated long video data could provide pseudo temporal grounding labels of specific questions, we utilize 30K QA pairs to warm up memory selection via a KL divergence loss. Subsequently, we use the rest 315K QA pairs to optimize the responses from the LLM and guide memory selection in a weakly-supervised manner. More training details are included in Appendix A.

## 4 Experiments

### 4.1 Datasets

We evaluate our model on long video QA datasets and present the statistics on the temporal duration of individual datasets in Table. 1. Among them, Next-QA [79], Next-GQA [80] and VideoChatGPT [50] encompass minute-long videos with thousands of frames. EgoSchema [51] contains over 5K three-minute videos with multiple-choice questions. Each question has a long temporal certificate, requiring more than 100 seconds within a video to produce a correct answer. MovieChat-1K [66] and MovieNet-QA [68] consist of around ten-minute-long or even hour-long movies, posing significant challenges for the model to comprehend the visual contents across such long time spans.

Table 3: Results on the fullset test split of EgoSchema [51].

| Method | Params | Fullset |
|---|---|---|
| *finetuned* | | |
| MC-ViT-L [7] | 424M | 44.4 |
| LongViViT [58] | 1B | 33.3 |
| *zero-shot* | | |
| InternVideo [73] | 478M | 32.1 |
| FrozenBiLM [84] | 890M | 26.9 |
| SeViLA [86] | 4B | 22.7 |
| LLoVi [87] | 7B | 33.5 |
| Vamos [72] | 13B | 36.7 |
| LangRepo [32] | 7B | 38.9 |
| LangRepo [32] | 8×7B | 41.2 |
| Ours | 7B+1.3B | **44.1** |

Table 4: Results on the validation set of Next-QA [79]. C, T, D denotes causal, temporal and descriptive splits.

| Method | Params | C | T | D | All |
|---|---|---|---|---|---|
| *finetuned* | | | | | |
| BLIP-2 [39] | 4B | 70.1 | 65.2 | 80.1 | 70.1 |
| LLaMA-VQA [35] | 7B | 72.7 | 69.2 | 75.8 | 72.0 |
| Vamos [72] | 7B | 72.6 | 69.6 | 78.0 | 72.5 |
| *zero-shot* | | | | | |
| InternVideo [73] | 478M | 43.4 | 48.0 | 65.1 | 49.1 |
| SeViLA [86] | 4B | 61.3 | 61.5 | 75.6 | 63.6 |
| Mistral [29] | 7B | 51.0 | 48.1 | 57.4 | 51.1 |
| LLoVi [87] | 7B | 55.6 | 47.9 | 63.2 | 54.3 |
| LangRepo [32] | 7B | 57.8 | 45.7 | 61.9 | 54.6 |
| LangRepo [32] | 8×7B | 64.4 | 51.4 | 69.1 | 60.9 |
| Ours | 7B+1.3B | **65.1** | **62.2** | **78.1** | **66.2** |

Table 5: Results on Next-GQA [80]. Acc@GQA is defined as the percentage of questions that are both correctly answered and visually grounded with IoP $\geq 0.5$.

| Method | Params | mIoP | IoP@0.5 | mIoU | mIoU@0.5 | Acc@GQA |
|---|---|---|---|---|---|---|
| *w/ specialized grounding module* | | | | | | |
| TempCLIP [59, 80] | 130M | 25.7 | 25.5 | 12.1 | 8.9 | 16.0 |
| SeViLA [86] | 4B | 29.5 | 22.9 | 21.7 | 13.8 | 16.6 |
| *w/o specialized grounding module* | | | | | | |
| LLoVi [87] | 7B | 20.7 | 20.5 | 8.7 | 6.0 | 11.2 |
| LangRepo [32] | 7B | 20.3 | 20.0 | 8.7 | 6.0 | 11.2 |
| LangRepo [32] | 8×7B | 31.3 | 28.7 | 18.5 | 12.2 | 17.1 |
| Ours | 7B+1.3B | **32.2** | **31.0** | **19.3** | **13.3** | **17.8** |

## 4.2 Main Results

In this section, we present the results of our 8.3B model (half of Phi-2 2.7B in streaming encoder and Vicuna-7B as the LLM). We omit the comparisons to proprietary LLMs.

**VideoChatGPT.** Table 2 presents the results on VideoChatGPT [50] in terms of Correctness of Information (CI), Detailed Orientation (DO), Contextual Understanding (CU), Temporal Understanding (TU) and Consistency (CO). Our model outperforms LLM-based video understanding methods on all five metrics, with a significant advantage in temporal understanding. It can be attributed to the memory-propagated streaming encoding architecture that explicitly captures temporal dynamics.

**EgoSchema.** In Table 3, we report the *zero-shot* performance on the fullset test split of EgoSchema [51]. MC-ViT [7] consolidates a long-term memory to memorize long contexts but requires finetuning on related dataset [19]. LLM-based methods [87, 32, 72] curate answers from the captions of segmented video clips. However, these short-term captions cannot be optimized end-to-end and inevitably lose some detailed information. In contrast, we use a trainable streaming encoder to produce memory embeddings in long videos and feed them into an LLM to generate responses. Our model outperforms all zero-shot methods and is comparable to the finetuned MC-ViT, demonstrating the effectiveness of our streaming architecture for long-term temporal modeling.

**Next-QA.** In Table 4, we perform *zero-shot* evaluation on the validation split of Next-QA [79] covering 5K multiple-choice questions. We respectively report the accuracy on Causal (C), Temporal (T) and Descriptive (D) subsets. Our method consistently surpasses all zero-shot counterparts. Typically, compared to LangRepo [32] with Mixtral-8×7B [30], our 8.3B model improves the causal, temporal, and descriptive accuracy by 0.7%, 10.8%, 9.0% with considerably fewer model parameters.

**Next-GQA.** Besides the evaluation of the generated responses, we also assess the temporal grounding ability on Next-GQA [80]. We calculate the Intersection of Prediction (IoP) and Intersection of Union (IoU), and use Acc@GQA to measure the accuracy of the correctly grounded predictions. According to the comparisons in Table 5, our simple similarity score based selection achieves the highest IoP and comparable IoU to SeViLA [86] with a specialized grounding module. Moreover, the highest Acc@GQA demonstrates the comprehensive capacity for grounding and high-level understanding.

**MovieChat-1K.** Table 6 shows the results on MovieChat-1K [66], including a global mode for overall long-term understanding and a breakpoint mode for detailed analysis of specific moments.

Table 6: Results on MovieChat-1K [66] global and breakpoint mode accuracy (Acc.) and score.

| Method | Global | | Breakpoint | |
|---|---|---|---|---|
| | Acc. | Score | Acc. | Score |
| VideoChat [42] | 57.8 | 3.00 | 46.1 | 2.29 |
| Video-LLaMA [88] | 51.7 | 2.67 | 39.1 | 2.04 |
| VideoChatGPT [50] | 47.6 | 2.55 | 48.0 | 2.45 |
| MovieChat [66] | 62.3 | 3.23 | 48.3 | 2.57 |
| MovieChat+ [67] | 71.2 | 3.51 | 49.6 | 2.62 |
| Ours | **90.4** | **4.42** | **54.9** | **2.80** |

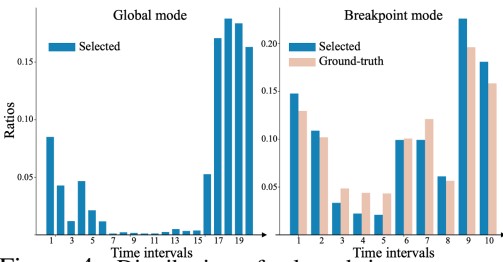

Figure 4: Distribution of selected timestamps on MovieChat-1K. We divide each video into multiple time intervals for statistical analysis.

Table 7: Results on MovieNet-QA [68]. We present the used modality, the average number of tokens input to LLM and the average inference latency per question for comprehensive comparison.

| Method | Text | Vision | Tokens | Latency | Overview | Plot | Temporal |
|---|---|---|---|---|---|---|---|
| LLaMA-VID [43] | ✓ | ✓ | 18430 | 16.03 sec | 3.09 | 3.31 | 2.02 |
| MovieLLM [68] | ✓ | ✓ | 18430 | 16.48 sec | 3.22 | 3.38 | 2.18 |
| LLaMA-VID [43] | ✗ | ✓ | 5477 | 10.47 sec | 2.28 | 2.88 | 1.46 |
| MovieLLM [68] | ✗ | ✓ | 5477 | 10.43 sec | 2.36 | 2.97 | 1.58 |
| Ours | ✗ | ✓ | 256 | 5.32 sec | **2.65** | **3.13** | **1.88** |

In breakpoint mode, [66, 67] manually extract segments according to the timestamps in questions, while our model adaptively selects the related historical memories. Fig. 4 reveals that our selected timestamps are close to the ground-truths, and the higher breakpoint accuracy validates our adaptive selection effectively gathers the desired information from long contexts. Meanwhile, we reach significantly superior results in global mode, with the model's selection concentrated at the beginning and ending parts. On the one hand, the beginning of a movie often contains hints of global information while the middle comprises redundant details. On the other hand, the condensed memories near the end of the video encapsulate the entire video, making them quite suitable for global understanding.

**MovieNet-QA.** Finally, we show the results on MovieNet-QA [68] consisting of 100 hour-long movies. Inspired by [68], we use GPT-3.5 to produce scores in range 0-5 to evaluate the performance in overview, plot, and temporal understanding in Table 7. Specifically, LLaMA-VID [43] compresses each frame into two tokens, which are then combined with movie subtitles as input to an LLM. MovieLLM [68] further incorporates more generated data in training. These approaches largely rely on the texts for movie understanding, and only using visual frames leads to dramatic performance drop. Moreover, its frame-wise compression is dependent on specific questions. The model has to reprocess the entire movie to extract visual features for different questions, resulting in a high inference latency of over 10 seconds per question. Conversely, our architecture requires only once streaming encoding to obtain a general condensed representation and adaptively selects significantly fewer tokens as input to LLM to answer specific questions. Therefore, we achieve a higher inference speed of 5.32 seconds per question and attain promising movie understanding without using subtitles.

**Qualitative Results.** We also present qualitative examples in Fig. 5. Typically, in Fig. 5a, our model accurately captures the detailed descriptions in the question, and precisely selects the relevant segments that contain the corresponding character. Moreover, in Fig. 5b, given a two-hour long movie and a high-level question on the movie plot, without relying on subtitles, VideoStreaming can comprehend the intent of the question and select relevant scenes from the lengthy video. In particular, the model selects the scenes of tightrope walk, team disputes, and equipment setup, clearly illustrating the protagonist's challenges, thereby contributing to a comprehensive answer generation.

### 4.3 Ablation Study

We first explore four different settings on the memory and selection design of our model in Table 8. The details are as follows: (1) No propogated memory and no temporal selection, the model degenerates into uniform sampling 64 frames for video understanding. (2) Without temporal selection, we adopt the memories of final 4 clips as the video-level representation. (3) Without propagated memory, we concatenate time indicators with questions to produce the question indicator to retrieve relative temporal segments. (4) The full architecture of VideoStreaming.

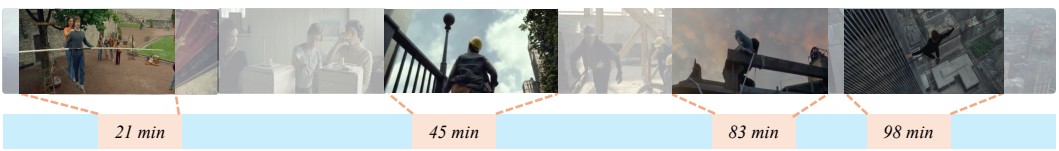

*User: Why is the man wearing white shirt snapping his fingers during the performance?*

| 0~22.5 sec | | 45.0~56.3 sec | | 78.8~90 sec |

*Assistant: The man in the white shirt is likely a member of the band and is playing an instrument or singing along with the group. He is snapping his fingers to keep the rhythm and add to the overall sound of the performance.*

(a) An example on Next-GQA. Our model accurately selects the segments containing the target character.

*User: What challenges does the protagonist in the movie face to achieve his goal?*

| 21 min | | 45 min | | 83 min | 98 min |

*Assistant: The protagonist in the movie confronts a series of obstacles in his endeavor to accomplish a remarkable feat. He must devise a method to access the towering structures, tackle the complexities of setting up the wire, maintain his equilibrium and concentration while suspended on a tightrope, and manage the risks associated with installing the equipment on the towers.*

(b) An example on a long movie. Our model selects typical segments that reveal the encountered challenges.

Figure 5: Examples of question answering and the selected timestamps based on specific instructions.

Table 8: Ablation studies on the effects of memory selection and historical memory in streaming encoding.

| Memory | Selection | Fullset | Global Acc. | Break. Acc. |
|--------|-----------|---------|-------------|-------------|
| ✗ | ✗ | 34.4 | 52.5 | 21.6 |
| ✓ | ✗ | 37.3 | 69.1 | 23.0 |
| ✗ | ✓ | 38.4 | 43.8 | 39.1 |
| ✓ | ✓ | 44.1 | 90.4 | 54.9 |

Table 9: Ablation studies on clip- and frame-based sampling strategy.

| Sampling | Metric | Fullset | MovieNet |
|----------|--------|---------|----------|
| Clip | Acc. | 44.1 | 2.56 |
| | Frames | 176 | 6032 |
| Frame | Acc. | 44.0 | 2.61 |
| | Frames | 180 | 6480 |

**Historical Memory.** In terms of the propagated memory in the streaming encoding process, i.e., $H_{k-1}$ in Eq 2, we report the fullset accuracy on EgoSchema [51] as well as global and breakpoint accuracy on MovieChat-1K [66] in Table 8. Typically, the historical memory significantly improves global understanding. This verifies our intuition that leveraging historical memory enables the model to produce a global representation that summarizes the entire video. Meanwhile, since we select a small portion of the encoded results from the long video as input to LLM, the propagated memory across clips increases tolerance for imperfect temporal selection, as it preserves previous contexts. Without memory, there would be a strict requirement on temporal selection accuracy to avoid completely losing necessary details, thus degrading the performance.

**Memory Selection.** We also validate the effects of our memory selection strategy. Comparing the results in Table. 8 with respect to temporal selection, we have two observations. First, for breakpoint mode needing detailed understanding of specific moments, the lack of temporal selection leads to dramatic performance drop. It is crucial to select the related clips otherwise the LLM cannot catch the necessary details. Second, the historical memories in streaming encoding process enable the encoded results from the final iterations to provide coarse summarization of the entire video. Whereas, as shown in Fig. 4, the beginning of the movie contains crucial cues for global understanding. Directly feeding the memories of final clips without temporal selection into LLM still results in information loss. These phenomena demonstrate the necessity of our adaptive selection for gathering detailed information over the long time span, which facilitates more accurate and informative responses.

**Sampling Strategy.** In default, we use clip-based sampling to segment a video into clips each consisting of a fixed number of frames. Alternatively, it is also feasible to directly sample at a moderate FPS, e.g., 1 FPS, and streamingly process the frames. Table 9 showcases the comparison between clip-based sampling and frame-based sampling at 1 FPS, and presents the average number of sampled frames on each dataset. The overall performance is close, which verifies that our model can well generailze to different scenarios. And on the longer MovieNet-QA videos, the frame-based sampling performs slightly better, as the clip-based approach limits the maximum number of clips, resulting in fewer total sampled frames for very long videos.

Table 10: Ablation studies on the streaming encoder architecture.

| Encoder | Layers | Params | Fullset | Next-QA | Global Acc. | Break Acc. |
|---------|--------|--------|---------|---------|-------------|------------|
| MC-ViT  | 24     | 0.4B   | 32.3    | 53.1    | 71.2        | 40.4       |
| Phi     | 4      | 0.3B   | 36.4    | 59.6    | 77.3        | 46.2       |
| Phi     | 8      | 0.7B   | 39.8    | 63.2    | 84.3        | 49.2       |
| Phi     | 12     | 1.0B   | 42.5    | 65.1    | 87.4        | 51.2       |
| Phi     | 16     | 1.3B   | 44.1    | 66.2    | 90.4        | 53.7       |
| Phi     | 24     | 2.0B   | 43.8    | 66.0    | 90.0        | 53.7       |
| Phi     | 32     | 2.7B   | 41.3    | 64.8    | 87.2        | 51.5       |
| Vicuna  | 3      | 0.7B   | 39.8    | 63.2    | 84.3        | 49.2       |
| Vicuna  | 6      | 1.3B   | 39.5    | 64.1    | 85.5        | 50.1       |
| Vicuna  | 12     | 2.7B   | 40.2    | 64.4    | 85.7        | 50.1       |

**Language Reasoning in Streaming Encoding.** Besides, we ablate the necessity of using a language model for memory consolidation. We compare our streaming encoder instantiated with partial layers of Phi-2.7B [20] with a conventional ViT-based approach, MC-ViT [7]. We compare the results on EgoSchema fullset [51] and Next-QA [79], Global and Breakpoint Accuracy on MovieChat-1K [66], as well as the number of encoder layers and parameters in Table 10. It is clear that using language reasoning to extract video memories exceeds the vision-based strategy even with fewer parameters. The reasons are twofold. On the one hand, MC-ViT updates memory according to handcrafted rules like clustering. In contrast, using language model as the streaming encoder allows us to leverage extensive video caption/QA data to guide memory consolidation in an end-to-end manner. This data-driven strategy results in more comprehensive memories that facilitate general video understanding. On the other hand, since the memory is fed into a subsequent LLM for reasoning, it is easier to align the memory generated by a language model with the input space of LLM, thus improving performance.

**Streaming Encoder Architecture.** Additionally, we also explore two alternative language models as the streaming encoder. (1) A comparatively a small language model, Phi-2.7B [20], with different partial layers. (2) The previous layers of the large language model, Vicuna-7B [13]. Interestingly, using comparatively fewer layers of the language model leads to better results. We conjecture this is because the language model is originally trained for next token prediction. Its feature space of the final Transformer decoder layer might not align with the objective of visual content condensation. Similar to [82, 46, 17], the shallower layers might produce feature embeddings that encode richer information and serve as more comprehensive condensed video representations. Besides, under the same parameters, the smaller language model retains more layers and consistently outperforms the previous layers of Vicuna. Meanwhile, the prefix task for training the streaming encoder requires first training the entire model for next token prediction, before using partial layers in later stages. Therefore, using Phi as the streaming encoder brings another advantage in reduced computation.

More ablation studies on temporal grounding supervision, the number of summarization tokens and selected timestamps, the time prompts, and the similarity measurement are included in Appendix D.

## 5   Conclusion

In this paper, we introduce a novel approach to tackle the complexities of long video understanding with large language models (LLMs). Our proposed memory-propagated streaming encoding architecture segments long videos into short clips and iteratively encodes each clip in sequence. By leveraging historical memory from preceding clips, we incorporate temporal dynamics into the encoding process and produce a fixed-length memory to encapsulate arbitrarily long videos. To further augment the detailed information for handling specific questions, we develop adaptive memory selection that selects relevant timestamps based on given instructions. This approach ensures that the most pertinent historical memories are utilized for question answering, thereby facilitating detailed and informative responses. Our model achieves superior performance with substantially fewer tokens and higher efficiency on extensive long video benchmarks. We demonstrate that memories from the streaming encoding significantly enhance global video understanding, while adaptive selection results in accurate temporal grounding with respect to specific questions.

## Acknowledgement

This project is funded in part by Shanghai Artificial Intelligence Laboratory, the National Key R&D Program of China (2022ZD0160201), the Hong Kong Generative AI Research and Development Center (HKGAI) under the Innovation and Technology Commission (ITC)'s InnoHK. Dahua Lin is a PI of HKGAI under the InnoHK.

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

## Limitations

One potential limitation is that we simply uniformly sample frames to form a set of short clips for memory-propagated streaming encoding. However, in a long video, different segments possess different amounts of information. The uniform sampling may result in using redundant tokens for clips with bland content. Meanwhile, the number of tokens used to represent clips with abundant visual contents and intensive temporal dynamics may be insufficient, leading to information loss. To address this limitation, we plan to explore adaptive segmentation techniques that dynamically adjust the segmented clip lengths based on the complexity and content of the video.

## Impact Statements

Our proposed VideoStreaming, a streaming long video understanding architecture with large language models has various potential impacts for society. On the positive aspect, VideoStreaming contributes to improved intelligent video understanding, especially for long videos. This could be beneficial in education, entertainment, and information retrieval, where users often need to navigate and understand complex video materials. Besides, our technique could lead to advancements in multimedia analytics with applications in areas like video surveillance, market research, and content personalization.

On the negative aspect, the ability to efficiently process and retrieve information from long videos raises potential privacy and security concerns. If misused, this technology could be employed for unauthorized surveillance, personal monitoring, or other unethical purposes that infringe on individual privacy. In addition, the enhanced video understanding capabilities might be exploited for the creation of manipulated or misleading video content, leading to the spread of misinformation and the potential for social manipulation.

In conclusion, despite that VideoStreaming presents advancement in long video comprehensive, its development should be accompanied by careful consideration of ethical and societal implications.

## A  More Implementation Details

We use CLIP ViT-L/14 [59] to extract frame-wise features with input resolution $224 \times 224$, resulting in 256 tokens per frame. Then, we concatenate every four spatially adjacent visual tokens along channel dimension, representing each frame with 64 tokens with channel dimension 4096. The streaming encoder consists of a two-layer MLP projector (channel dimension 4096-2560-2560) with GELU activation [23] and a language model Phi-2 2.7B [20]. In the first training stage, we initially freeze Phi-2, and only tune the MLP projector on 790K caption pairs, including 558K image caption data from CC3M [64] and 232K short video caption data from WebVid 2.5M [6]. Following LLaVA [46, 45], we use AdamW optimizer [47] with global batchsize 256, initial learning rate $1 \times 10^{-3}$ with cosine decay to train 1 epoch for modality alignment. Subsequently, we jointly train Phi-2 and the MLP projector on 763K QA pairs, including 625K image QA pairs [18, 26, 33, 37, 46, 52, 53, 63, 65], 40K text conversations [1] and 98K video QA pairs [10], with global batchsize 128, initial learning rate $2 \times 10^{-5}$ with cosine decay.

In the second stage, we jointly train ViT, the streaming encoder and the LLM on long video data. In the memory-propagated streaming encoding process, we insert a brief prompt to indicate the explicit timestamps of the historical memory and the input clip formulated as *This contains a history of {start} to {end} seconds, and a clip sampled in {start} to {end} seconds.*. We adopt the output of the first 16 layers out of the 32 layers of Phi-2 as the condensed representation. Then, we adaptively select 4 most relevant timestamps and feed the associated 256 memory tokens into a two-layer MLP projector with channel dimension 2560-4096-4096 and an LLM, Vicuna-7B [13] to generate the final responses. We jointly train the whole architecture, including Vicuna, Phi-2, MLP projectors and ViT encoder, on long video QA data with global batchsize 128, initial learning rate $2 \times 10^{-5}$ with cosine decay. In default, we first use 20K synthesized long videos and sample 10K QA pairs curated from Panda-70M with pseudeo temporal grounding labels to train memory selection as warm-up. The learning objectives contain a standard next token prediction loss and a supervised KL divergence loss that aligns the distribution of the predicted memory selection results and the pseudo temporal labels. Next, based on the warm-up model, we further train on the rest 295 long video QA pairs only with

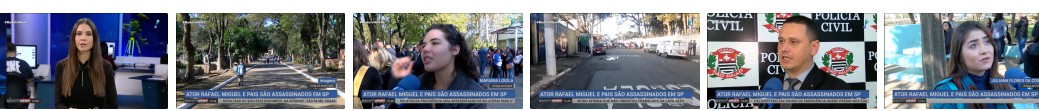

*[1, 12]: The speaker is a female news reporter who is sitting in front of a studio camera.*

*A: In 1 to 12 seconds, the speaker is a female news reporter who is sitting in front of a studio camera. In 15 to 23 seconds, a group of people standing in line outside of a building. In 25 to 28 seconds, two children standing next to each other on a tv screen. ⋯ In 162 to 166 seconds, a woman is interviewed by a reporter in a park.*

*Q: What is the timestamp that a man in a suit and tie is talking into a microphone at a press conference?*

*A: A picture of a person and their family from the news is featured in a video.*

Figure 6: An example of the QA pairs from the captions and segmented timestamps from Panda-70M [11].

Table 11: Results on the test set of IntentQA [41].

| Method | Params | Why | How | Before/After | All |
|---|---|---|---|---|---|
| LLaMA-VID [43] | 13B | 43.8 | 40.1 | 36.3 | 41.4 |
| LLoVi [87] | 7B | 57.9 | 55.4 | 42.3 | 53.6 |
| LangRepo [32] | 7B | 56.9 | 60.2 | 42.1 | 53.8 |
| LangRepo [32] | 8×7B | 62.8 | 62.4 | 47.8 | 59.1 |
| Ours | 7B+1.3B | **65.6** | **66.2** | **59.0** | **64.1** |

next token prediction loss. The whole training is conducted on 32 A100 (80G) GPUs for around 2.5 days.

## B  Long Video QA Data Creation

In addition to the existing 25K long video QA pairs on movies [66, 43], we create more QA data from two aspects. First, we leverage the existing short video QA dataset [76, 81] and synthesize short videos into minute-long videos with average duration of one minute. The original questions of each short video coarsely correspond to a temporal segment in the synthesized long video. We use this correspondence as noisy labels to supervise the memory selection. Second, recent Panda-70M [11] segments long videos into short clips and produces captions for each clips. This dataset provides the original long videos, the captions of segmented clips as well as the segmentation timestamps. Based on these cues, we produce multi-round QA conversations. Below we show an example in Fig. 6. The produced time-sensitive QA pairs are crucial to enhance the temporal awareness and guide precise memory selection in long videos.

## C  More Experimental Analysis

**Quantitative Results on IntentQA.** IntentQA [41] is a long-form video understanding dataset consisting of 4.3K videos with 16K multiple-choice questions, which are classified into three types, why, how and before/after. In Table 11, we report the zero-shot performance on IntentQA test set. Our method presents a dominant advantage in temporal understanding. And the overall performance is significantly superior to recent works.

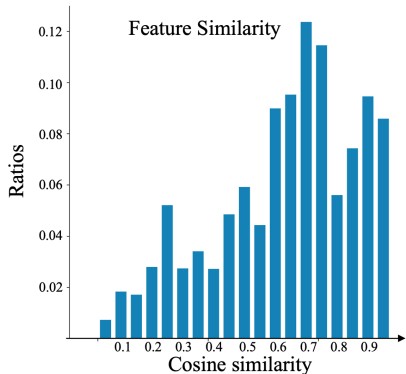
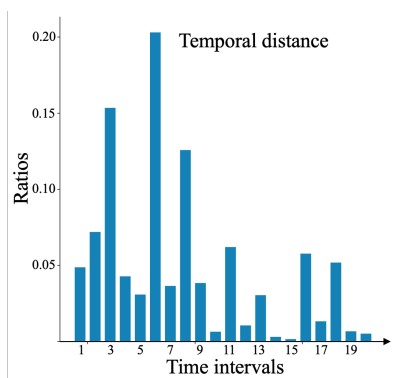

Figure 7: Visualization of the feature similarity and temporal distance of selected clips.

Table 12: Ablation studies on the use of temporal grounding supervision. Acc denotes the ratio of correctly answered questions on Next-GQA [80] regardless of grounding accuracy.

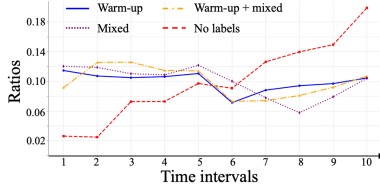

| Warm-up | Mixed | Fullset | mIoP | mIoU | Acc@GQA | Acc |
|---------|-------|---------|------|------|---------|-----|
| ✗ | ✗ | 43.7 | 24.1 | 9.8 | 11.1 | 54.9 |
| ✓ | ✗ | 44.1 | 32.2 | 19.3 | 17.8 | 55.7 |
| ✗ | ✓ | 43.9 | 28.5 | 14.6 | 15.4 | 55.3 |
| ✓ | ✓ | 44.0 | 32.1 | 20.0 | 17.7 | 54.8 |

Figure 8: Distribution of selected timestamps on Next-GQA.

**Analysis on the Selected Clips.** To verify whether the selected clips are redundant, we provide statistics on the feature similarity and the temporal distribution of the selected clips in Fig. 7. For feature similarity, we calculate the cosine similarity of the time indicators of the selected clips. We divide the x-axis from 0 to 1 into 20 bins, and it shows the distribution of feature similarity. The average cosine similarity is 0.68. For temporal distance, we calculate the time intervals between the selected clips (represented as the ratio of the total video duration) and visualize the distribution of these time distances. The average distance is around 35% of the total video length. The statistics on the feature similarity and temporal distance indicate that the selected clips are not redundant.

## D  More Ablation Studies

We provide more ablation studies on the use of temporal grounding supervision, the number of summarization tokens and selected timestamps, the effects of time prompts in the memory-propagated streaming encoding process, and the similarity measurement used in memory selection.

**Temporal Grounding Supervision.** First, we present the studies on the use of temporal grounding supervision. As mentioned in Section 3.4, we employ around one-tenth of long video QA pairs to provide pseudo temporal labels. We compare four training strategies: (1) Fully weakly-supervised manner without any pseudo labels. (2) Using pseudo labels to train a *warm-up* model, then expanding to large-scale QA pairs. (3) *Mixing* all long video QA data, where the model uniformly receives temporal supervision in training. (4) Training on mixed data after warm-up initialization. The results on EgoSchema [51] and Next-GQA [80] in Table 12 indicate three key points: First, warm-up training contributes to more powerful grounding ability. The sparse temporal label supervision in mixed mode is overcome by the powerful initialization from warm-up training, which can generalize to large-scale data. Second, reusing the temporal labels after warm-up offers no additional benefits, so we adopt warm-up as the default setting. Third, without using temporal labels, the grounding performance drops, but the QA accuracy remains stable. Fig. 8 reveals that compared to those trained with temporal labels, the weakly-supervised model selects relatively later segments that preserve previous contexts with the help of historical memory, thus maintaining comparable QA capacity.

**The Number of Summarization Tokens and Selected Timestamps.** We compare using different number of summarization tokens and selected timestamps, i.e., $P$ in Eq. 1 and $V$ in Eq. 3. We

Table 13: Ablatoin study on the number of summarization tokens and selected timestamps. We report the results on EgoSchema [51] and Next-GQA [80].

| $P$ | $V$ | Tokens | Fullset | Acc@GQA |
|-----|-----|--------|---------|---------|
| 1 | 4 | 64 | 32.1 | 9.8 |
| 1 | 8 | 128 | 33.4 | 10.5 |
| 4 | 1 | 64 | 41.6 | 15.5 |
| 4 | 4 | 256 | 44.1 | 17.8 |
| 4 | 8 | 512 | 44.9 | 18.0 |
| 16 | 1 | 256 | 42.5 | 16.3 |
| 16 | 4 | 1024 | 43.8 | 17.9 |

Table 14: Ablation study on the formulation of time prompts. We report the results on EgoSchema [51], Next-GQA [80] and MovieChat-1K [66].

| Prompt | Fullset | Acc@GQA | Global Accuracy | Breakpoint Accuracy |
|--------|---------|---------|-----------------|---------------------|
| None | 38.8 | 12.4 | 66.7 | 19.8 |
| Clip | 40.5 | 15.4 | 70.3 | 44.5 |
| Memory | 42.1 | 16.1 | 83.3 | 43.1 |
| Clip+Memory | 44.1 | 17.8 | 90.4 | 54.9 |

compare the performance as well as the number of tokens input to LLM in Table 13. We conclude three observations. First, too few summarization tokens, e.g., $P = 1$, leads to substantial performance drop, since it condenses a 16-frame into only 16 tokens with significant information loss in spatial contexts. Such information loss cannot be compensated by selecting more temporal segments. Second, the performance saturates when improving $P$ from 4 to 16. This is because the existing video benchmarks [51, 80] do not place high demands on spatial detail understanding. It is sufficient to represent each frame with 4 tokens on average. Third, increasing the number of selected timestamps only results in minor improvements, which is not proportional to the increased number of tokens. This can be attributed to the historical memory used in the streaming encoding process. The utilization of historical memory enables the condensed representation of each clip to encompass the information in preceding clips, which enlarges the temporal receptive field. Hence, increasing the number of selected timestamps does not proportionally increase the temporal receptive field, resulting in slight performance improvements.

**Time Prompts.** We explore three different formulations of the time prompts used in memory-propagated streaming encoding: (1) Only with the timestamps of the current clip, e.g., *This clip is sampled in {start} to {end} seconds.* (2) Only with the timestamps of the historical memory, e.g., *This contains a history of {start} to {end} seconds.* (3) Simultaneously with the timestamps of the historical memory and the current clip, e.g., *This contains a history of {start} to {end} seconds, and a clip sampled in {start} to {end} seconds.* We report the results of different time prompts in Table 14. It is obvious that the lack of time prompts leads to substantial performance drop in the MovieNet-1K breakpoint mode accuracy, which requires detailed analysis of specific moments. The reason is that the breakpoint mode requires the model to answer questions at specified timestamps, the time prompts provide the model with necessary information in adaptive selection. Meanwhile, incorporating the timestamps of historical memories results in more significant improvements in global understanding. Overall, jointly leveraging the memory and clip timestamps contributes to the best results.

Table 15: Ablation study on the similarity measurement. We report the results on EgoSchema [51], Next-GQA [80] and MovieChat-1K [66]

| Similarity | Fullset | Acc@GQA | Global Accuracy | Breakpoint Accuracy |
|------------|---------|---------|-----------------|---------------------|
| Cosine | 44.1 | 17.8 | 90.4 | 54.9 |
| Dot product | 34.5 | 9.3 | 55.6 | 22.1 |

Table 16: Ablation study of different settings, including memory propagation, temporal selection, the number of summarization tokens and the number of selected clips, on hour-long MovieNet-QA benchmark from three perspectives.

| Memory | Selection | $P$ | $V$ | Overview | Plot | Temporal |
|:---:|:---:|:---:|:---:|:---:|:---:|:---:|
| ✗ | ✗ | - | - | 1.98 | 2.23 | 1.39 |
| ✓ | ✗ | 16 | 4 | 2.51 | 2.61 | 1.63 |
| ✗ | ✓ | 16 | 4 | 2.24 | 2.77 | 1.52 |
| ✓ | ✓ | 16 | 4 | 2.65 | 3.13 | 1.88 |
| ✓ | ✓ | 4 | 4 | 2.53 | 2.82 | 1.73 |
| ✓ | ✓ | 4 | 8 | 2.58 | 3.02 | 1.83 |
| ✓ | ✓ | 16 | 8 | 2.68 | 3.17 | 1.95 |

**Similarity Measurement.** Besides, we present the study on the similarity measurement used in adaptive memory selection. We compare the default cosine similarity against simple dot product without normalization in Table 15. Empirically, we observe that dot product could result in numerical instability, leading to overflow in training. Consequently, the calculated similarity score cannot reflect the correlation between the instruction and different segments and results in poor results on questions that require accurate temporal grounding, e.g., the Acc@GQA metric on Next-GQA [80] and the breakpoint mode accuracy on MovieChat-1K [66].

**Analysis on Long Video Benchmark.** Finally, we show the ablation studies of different settings particularly on the hour-long video benchmak, MovieNet-QA in Table 16. It is clear that for hour-long videos, the propagated memory is crucial for understanding the overview contents, detailed plots as well as temporal relations. Meanwhile, the temporal selection significantly improves detailed plot analysis, and appropriately increasing the number of selected clips can further improve performance.

