# OpenReview forum: "Streaming Long Video Understanding with Large Language Models"
_NeurIPS.cc/2024/Conference — NeurIPS 2024 poster_

### Official Review · Reviewer_sqPz · 2024-07-12

**Soundness:** 3
**Presentation:** 3
**Contribution:** 3
**Rating:** 6
**Confidence:** 4

**Summary:**

This paper proposes VideoStreaming to understand arbitrary-length videos by streamingly encoding video tokens and adaptively selecting a constant number of them. Through careful designs, including a small LLM streaming encoder, prefix task, modified attention, memory propagation, and Gumbel-Topk token selection, the method successfully achieves superior performance on long video benchmarks, showcasing precise temporal comprehension for detailed question answering.

**Strengths:**

1. This paper has a strong motivation, aiming to solve the significant computational burden caused by the large number of tokens extracted from long videos. This is an important issue for LMM to understand long videos.

2. The methodology is novel. First, the paper utilizes the causal LLM to satisfy the streaming and query-related vision summarization with explicit timestamp cues. Second, the paper designs a prefix task that naturally compresses the vision tokens during auto-regressive training. Lastly, through memory propagation and adaptive selection, the model successfully uses only the contextually related tokens to forward to the LLM, thus reducing the computational and memory cost due to the long kv cache.

3. The paper is well written and easy to follow.

**Weaknesses:**

My main concerns are as follows:

A. In common short/medium video length benchmarks, VideoStreaming could be unnecessary;

B. For long video length benchmarks, the paper does not provide enough experiments;

C. Promising experiments, like real-time interaction, are completely ignored in this paper.

Specifically,

A.1. This paper actually uses much more data during training compared to LLaVA-Next-Video:

- LLaVA-Next-Video-7B: 558K for MLP + 860K for MLP & LLM

- VideoStreaming-8.3B: 790K for MLP + 763K for small LLM + 345K for all

VideoStreaming is trained with 480K more data, which cannot be ignored. However, note that LLaVA-Next-Video-7B has slightly better results on VideoChat-GPT and even employs a much simpler pooling strategy to encode video tokens. Given that VideoChat-GPT's average duration is 1.81 minutes (see Table 1), we can infer that when the short video is less than 2 minutes, VideoStreaming may not offer advantages.

A.2. For medium-length videos, such as EgoSchema (average 3 minutes), the paper seems to have missed the results of VideoChat2 mistral-7B, which shows an accuracy of 54.4%, much higher than the 44.1% reported in the paper. Although it is not entirely fair since VideoChat2 mistral-7B uses even more training data, its model employs a simple Q-former with 16 segments and 16 frames to handle the video. Therefore, I also suspect that VideoStreaming may not present advantages in medium-length videos. Here I have omitted the discussion of MovieChat-1K since its question setting does not really relate well to minute-long videos.

B. Therefore, where VideoStreaming might truly excel is in handling very long videos. However, it is very disappointing that the paper does not present enough experiments on this. The ablations have been done on medium-length videos, and the authors do not provide any competitive baselines (with the same training data and procedure) in Table 7.

C. In fact, VideoStreaming completely ignores a promising application, which is long-term, real-time streaming QA, such as GPT-4o (concept). The method can significantly reduce the kv cache consumption during long video streaming and maintain high-speed response even if a long sequence has been input.

D. The demo shown in Figure 5(b) can be misleading. If you try the text-only Vicuna 13b (https://chat.lmsys.org/) and ask, "What challenges does Philippe Petit face in his quest to walk between the Twin Towers?" it can produce even better responses, such as "getting access to the top of the Twin Towers," "creating the perfect wire," "practicing the walk," and "keeping the walk a secret." This suggests that the question you provided may only prompt the LLM to recall its text pretraining data. I also doubt that the model can precisely select the relevant clips in a 2-hour movie, as that would be an extraordinary feat even for GPT-4o or Gemini-1.5-Pro.

Despite the above weaknesses, I still acknowledge the novelty of the methodology. I will decide on my final rating after reading the rebuttal.

**Questions:**

1. Would it be possible to use the previous layers of Vicuna to replace the extra introduced (half) Phi-2, which has 1.3B parameters that cannot be ignored?

2. Should L127-L129 be correct? StreamingLLM [a] has demonstrated the importance of initial tokens. I think it is your designed prefix task that restricts the streaming encoder from achieving this.

3. The method still needs to segment clips. Why not directly operate on the frames with a moderate FPS (e.g., 2)? This should make the method more general. Or will this also incur a huge memory cost for the streaming encoder since it is still at the billion-level?

4. The ablation in Table 8 is hard for me to understand. First, what is the performance if there is no memory and no selection? Second, the performance variance in MovieChat is so huge. Are these results from different trained models, or just the same trained model but different inference?

[a] Efficient Streaming Language Models with Attention Sinks, ICLR 2024.

**Limitations:**

The limitations are discussed in the supplementary material: (1) simply uniformly sampling frames to form a set of short clips for memory-propagated streaming encoding rather than using adaptive segmentation; (2) the number of tokens may not be sufficient to represent clips with abundant visual content and intensive temporal dynamics. These are indeed limitations and are well discussed. However, as mentioned in the 'weakness' section, most limitations pertain to the experimental setting.

---

> ### Author Rebuttal · Authors · 2024-08-07
>
> We sincerely thank the reviewer for recognizing our novelty, method and paper writing. Below we provide point-to-point responses to all the raised questions:
>
> > **Q1: The necessity of VideoStreaming on short-term videos.**
>
> **A1:** For short videos, sampling multiple frames and treating it as a multi-image understanding task works well. However, this is not suitable for long videos due to excessive number of frames. As we focus on long-form videos, the short video results are mainly to demonstrate our method also performs well in that setting. Since LLaVA-Next-Video is a concurrent work updated on April 30th, we did not include a comparison.
>
> > **Q2: The results on medium-length videos.**
>
> **A2:** The results of VideoChat2 Mistral-7B on EgoSchema is updated on May 23rd (in arxiv:2311.17005 v4), which is later than the submission. Technically, the accuracy of 54.4\% is obtained by using 2M SFT data including ego-centric QA and more powerful LLM. Since it is difficult to obtain the same training data in a short period of time, we train VideoStreaming with Mistral-7B on our own data and compare the results on three benchmarks in Table 4 in rebuttal PDF.
> + On EgoSchema, we achieve 48.1\%. The gap to VideoChat2 is majorly due to the lack of ego-centric training data.
> + On recent popular MLVU and VideoMME benchmarks consisting of medium and long videos, our model significantly outperforms VideoChat2 by 7.2 on MLVU multi-choice questions, 6.1 and 6.4 on VideoMME medium and long subset respectively.
>
> > **Q3: More experiments on long videos and explanation of Table 8.**
>
> **A3:** In rebuttal PDF, we provide more ablations on long videos in Table 6 and add explanations for different settings in Table.
>
> In Table 6, we ablate the memory propagation, temporal selection, number of summarization tokens $P$ and number of selected clips $V$ on MovieNet-QA.
> + For hour-long videos, the propagated memory is crucial for understanding from all aspects.
> + The temporal selection significantly improves detailed plot analysis, and appropriately increasing the number of selected clips can further improve performance.
>
> For the performance variance on MovieChat, the reasons are below.
> + For global mode requiring long-term comprehension, the lack of memory will cause the LLM to have very limited temporal receptive fields and lead to dramatic performance drop.
> + Besides, as shown in Figure 4 in manuscript, the beginning of the movie contains crucial cues for global understanding. Directly feeding the memories of final 4 clips without temporal selection into LLM also results in information loss.
> + For breakpoint mode needing detailed understanding of specific moments, it is important to select the related clips otherwise the LLM cannot catch the necessary details.
> + Further, the propagated memory across clips increases tolerance for imperfect temporal selection, as it preserves previous contexts. Without memory, there would be a strict requirement on temporal selection accuracy to avoid completely losing necessary details.
>
> > **Q4: The promising application like GPT-4o.**
>
> **A4:** Yes, VideoStreaming can disentangle video encoding and question-answering into two asynchronous processes for streaming QA with low time latency. We are working on the deployment to make it a more promising application. We did not report it in the paper due to the lack of suitable benchmarks for evaluation.
>
> > **Q5: About the example in Figure 5.**
>
> **A5:** Thanks for pointing out this. We find that some specialized words like 'Philippe Petit' and 'Twin Tower' can make LLM recall the text pretraining data. To solve this problem, we replace the words that may cause information leakage and formulate a new question as shown in Figure 1 in rebuttal PDF. The comparison between VideoStreaming and text-only LLM demonstrates that our model can comprehend the high-level expressions in the question and summarize the relative contents in the visual contexts.
>
> > **Q6: The grounding ability in Figure 5.**
>
> **A6:** We use a similarity-based strategy to select the clips related to specific questions, which relies on some semantic relations to do selection. Even with an implicit question expression, our model can correlate the scenes that contain the protagonist with the question as shown in Figure 1 in rebuttal PDF.
>
> > **Q7: Use previous layers of Vicuna to replace Phi.**
>
> **A7:** Using previous layers of Vicuna to replace half Phi will cause two problems.
> + Our streaming encoder is trained with a prefix task to generate outputs that only reference memory tokens to build a robust memory. This requires first training the entire model for next token prediction, before using partial layers in later stages. Replacing Phi with Vicuna layers forces us to first train the full 7B model, significantly increasing computational cost.
> + If we take the previous few layers of Vicuna and maintain the same parameters with Phi, we can only retain a very small number of layers. Table 5 in rebuttal PDF shows the performance drops when replacing Phi with previous Vicuna layers under the same parameters.
>
> > **Q8: The importance of initial tokens.**
>
> **A8:** Yes, you are correct. It is our prefix task that ensures the information of a sequence is distilled into the last few tokens. We will accordingly modify the presentation.
>
> > **Q9: Operate on the frames instead of clips.**
>
> **A9:** Yes, our model can be applied to the scenario by sampling frames at a moderate FPS. In Table 7 in rebuttal PDF, we show the comparison between clip-based sampling and frame-based sampling at 1 FPS, and present the average number of sampled frames on each dataset.
> + The overall performance is close since the two sampling strategies lead to comparable number of sampled frames.
> + On the longer MovieNet-QA videos, the frame-based sampling performs slightly better, as the clip-based approach limits the maximum number of clips, resulting in fewer total sampled frames for very long videos.

---

> ### Author Response · Authors · 2024-08-12
> **Additional Response to Reviewer sqPz**
>
> Dear Reviewer sqPz,
>
> We sincerely thank you again for your great efforts in reviewing this paper, especially for the precious advice that has helped us improve the quality of this paper significantly!
>
> In response to your suggestions, we have supplemented the experiments and discussions in the rebuttal. If there are any further clarifications or additional experiments you would like us to address, please do not hesitate to let us know.
>
> Best,
>
> Paper 6273 Authors.

---

> ### Comment · Reviewer_sqPz · 2024-08-12
> **Many thanks for the detailed response**
>
> > A1: ...
>
> The response somehow works for me. Let's focus more on the long-form videos. However, I do think that in a future version, you should provide an ablation study on short-term videos to show whether your method decreases performance compared to the simple pooling strategy.
>
> > A2 & A3: ...
>
> Nice & important results. Many thanks for that.
>
> > A4: ...
>
> The recommendation is to validate on some online/streaming action detection benchmarks, or follow Streaming Vid2Seq [a]/VideoLLM-online [b] on CVPR 2024.
>
> [a] Streaming Dense Video Captioning. arXiv: 2404.01297.
>
> [b] VideoLLM-online: Online Video Large Language Model for Streaming Video. arXiv: 2406.11816.
>
> > A5: ...
>
> Now the example is less misleading. Thanks for that. However, I wonder if there are instances, such as a movie title at the beginning frame, where knowledge could still be leaked. Could you provide non-movie examples? If the rebuttal cannot provide an anonymous link, please include them in a future version.
>
> > A6: ...
>
> This does not convince me. I suspect the grounding is cherry-picked and there are many false positives and true negatives. However, I understand that this is what the demo needs.
>
> > A7 & A9: ...
>
> Thanks for that. The results are consistent with my estimate.
>
> Overall, I appreciate the hard work of the authors. Most of my concerns have been addressed, and I plan to increase my rating score.

---

> > ### Author Response · Authors · 2024-08-13
> >
> > Dear Reviewer sqPz:
> >
> > We are delighted to hear that most of your concerns are addressed! We will include the ablation study on short-term videos, the experiments on online/streaming benchmarks, and non-movie examples in the revised version.
> >
> > Regarding your concerns in A5 and A6, there is no movie title at the beginning frame, only the visual tokens and question tokens are input to the LLM to produce the answer. For grounding, there exist reasonable alternatives to be selected in the movie, but the propagated memory allows the model to select a subset to answer the questions. We will present more detailed grounding results in the revised version.
> >
> > Many thanks again for your very constructive comments, which have helped us improve the quality of the paper significantly.
> >
> > Best,
> >
> > Paper 6273 Authors.

---

### Official Review · Reviewer_9Ye5 · 2024-07-13

**Soundness:** 3
**Presentation:** 3
**Contribution:** 3
**Rating:** 6
**Confidence:** 4

**Summary:**

This paper introduces VideoStreaming, a Vision-Language Large Model (VLLM) designed for comprehensive video understanding. It effectively manages the challenges of computational overhead associated with processing long videos by innovatively using a constant number of video tokens. VideoStreaming demonstrates superior performance and efficiency on long video benchmarks, excelling in detailed question answering with accurate temporal comprehension.

**Strengths:**

1.The Memory-Propagated Streaming Encoding technique effectively captures long-term video content while maintaining a manageable computational footprint by propagating memory through sequential segments.
2. Adaptive Memory Selection strategically reduces the number of tokens used in processing, selecting only the most relevant memories for generating responses, which enhances the model's efficiency and precision.
3. Demonstrated superior performance on long video benchmarks indicates the model's capability to handle complex video understanding tasks effectively, setting it apart from existing models.

**Weaknesses:**

1. The core strategy still relies on processing video through key clips, which missed some relevant works, such as "LGDN: Language-guided denoising network for video-language modeling" (NeurIPS 2022).
2. The method for selecting key clips could potentially lead to redundancy, especially if multiple similar clips have high relevance scores. Integrating a regularization function to promote diversity in key clip selection could help mitigate this issue.
3. In scenarios involving multiple queries about a video, the model may need to repeatedly extract key clips for each question, potentially affecting efficiency. The paper could benefit from a discussion on optimizations that reduce the computational overhead in such use cases.

**Questions:**

Please see weaknesses. I will raise my score if the author well address my concerns.

**Limitations:**

None.

---

> ### Author Rebuttal · Authors · 2024-08-07
>
> We sincerely thank the reviewer for acknowledging our model's ability to handle long videos with high efficiency. Below we would like to provide point-to-point responses to all the raised questions:
>
> > **Q1: Related work on processing key clips.**
>
> **A1:** Thanks for pointing out this related work. LGDN is a pioneering work in selecting the key clips that are highly related to the text prompts. It employs a contrastive learning architecture to learn global video-text correspondence and then uses the model to calculate fine-grained clip-text correspondence scores to filter out noisy frames. The key difference is that we can leverage both fine-grained temporal grounding labels and implicit gradient flow from the next token prediction loss to guide the learning of temporal selection. We will add the discussions in the related work.
>
> > **Q2: The redundancy and diversity of selected clips.**
>
> **A2:** Thanks for the promising suggestion of integrating a regularization term in the key clip selection stage. In the author rebuttal PDF, we provide statistics on the feature similarity and the temporal distribution of the selected clips in Figure 2.
> + For feature similarity, we calculate the cosine similarity of the time indicators of the selected clips. We divide the x-axis from 0 to 1 into 20 bins, and it shows the distribution of feature similarity. The average cosine similarity is 0.68.
> + For temporal distance, we calculate the time intervals between the selected clips (represented as the ratio of the total video duration) and visualize the distribution of these time distances. The average distance is around 35% of the total video length. The statistics on the feature similarity and temporal distance indicate that the selected clips are not redundant.
>
> We will explore integrating a regularization term in future work.
>
> > **Q3: The efficiency of key clip extraction with multiple queries.**
>
> **A3:** It is worth noting that in our architecture, the video encoding process and the question-answering process are disentangled. Our model only requires once forward computation to encode a video into memories, which are independent of specific questions. When encountering multiple questions for a video, we only need to calculate the similarity between multiple question indicators and the obtained time indicators to select different key clips. This process requires much fewer computations than the streaming video encoding and is efficient.

---

> ### Author Response · Authors · 2024-08-12
> **Additional Response to Reviewer 9Ye5**
>
> Dear Reviewer 9Ye5,
>
> We sincerely thank you again for your great efforts in reviewing this paper, especially for the precious advice that has helped us improve the quality of this paper significantly!
>
> In response to your suggestions, we have supplemented the discussions and clarifications in the rebuttal. If there are any further clarifications or additional experiments you would like us to address, please do not hesitate to let us know.
>
> Best,
>
> Paper 6273 Authors.

---

> > ### Comment · Reviewer_9Ye5 · 2024-08-13
> >
> > Thank you for the detailed responses. My concerns have been addressed, and I would like to raise my score.

---

> > > ### Author Response · Authors · 2024-08-13
> > >
> > > Dear Reviewer 9Ye5:
> > >
> > > We are delighted to hear that your concerns are addressed! Many thanks again for your very constructive comments, which have helped us improve the quality of the paper significantly.
> > >
> > > Best,
> > >
> > > Paper 6273 Authors.

---

### Official Review · Reviewer_QjSL · 2024-07-15

**Soundness:** 3
**Presentation:** 3
**Contribution:** 3
**Rating:** 5
**Confidence:** 4

**Summary:**

This paper presents an advanced vision-language model designed to handle the complexities of understanding arbitrary-length videos. The model addresses the computational challenges posed by long video sequences by implementing two core techniques: Memory-Propagated Streaming Encoding and Adaptive Memory Selection. These innovations enable the model to efficiently encode long videos into a constant number of video tokens, preserving both spatial and temporal dynamics. Extensive experiments demonstrate that VideoStreaming achieves superior performance and higher efficiency on long video benchmarks, showcasing its ability to provide precise temporal comprehension for detailed question answering. The contributions include a novel approach to long video understanding, efficient encoding and memory selection strategies, and comprehensive evaluations that highlight the model’s capabilities and advantages over existing methods.

**Strengths:**

- The paper addresses a significant and challenging research problem: long video understanding.
- The proposed strategy is technically sound.
- The paper conducts sufficient experiments on varied benchmarks, utilizing six video datasets of different durations.
- The use of an additional small encoder, such as Phi-2, as a streaming encoder is an interesting approach.

**Weaknesses:**

- Summarization Tokens: The use of summarization tokens is unclear, as it avoids attention between the TT tokens and the clip TN tokens. This design choice needs better justification and clarification.

- Small Language Model for Streaming Encoder: While the use of a small language model for the streaming encoder is interesting, there are no experiments demonstrating the necessity of language reasoning for visual-dominant dependencies (mainly visual embed. outputs). Additionally, the model is still quite large (2.7 billion parameters). A more efficient alternative, such as a shadow layer transformer, should be considered.

- Paper Writing: The proposed technical strategy is difficult to understand and lacks readability. It would benefit from clearer explanations and more detailed descriptions.

- Memory-Propagated Strategy: The memory-propagated strategy is similar to previous strategies in video understanding, such as those in [1]. The paper should differentiate its approach more clearly.

[1] Memory-Augmented Dense Predictive Coding for Video Representation Learning

- Ablation Studies: The ablation studies are not comprehensive, with only Table 8 presented. There is a lack of convincing studies on several designs, such as summarization tokens. More detailed ablation studies are needed to validate the design choices.

- Baseline Comparisons: In the main tables, the paper does not include recent instruction-tuning baselines such as LLaMA-VID. Many baselines, like LangRepo and LLoVi, are mainly modular methods based on image captioners like LLaVA, making the comparisons unfair. Including more relevant and recent baselines would provide a clearer picture of the paper's contributions.

- Progressive Training: The progressive training strategy is not innovative and can be found in previous works [1, 2]. The paper should highlight any unique aspects of its approach.
[1] LLaMA-VID
[2] A Simple Recipe for Contrastively Pre-training Video-first Encoders Beyond 16 Frames

**Questions:**

- Performance on Egoschema: I am very curious about the performance on the Egoschema dataset. As it is an egocentric video dataset, and since the pretraining dataset does not include egocentric videos, how does the model achieve a significant zero-shot performance (44.1) comparable to the finetuned MC-ViT-L baseline (44.4)? Could you provide more insights or experiments that explain this result?

- Visualization in Figure 5(b): The accurate grounding along a video exceeding one hour is impressive. However, I wonder if the LLM already recognizes specific information from the query, such as "Philippe Petit" and "Twin Towers." Additionally, the grounding intervals and the text outputs do not seem to have solid associations. Could you clarify how the grounding intervals are determined and ensure that they are directly linked to the text outputs? Are there any qualitative or quantitative measures used to validate these associations?

**Limitations:**

Please refer to the weakness & questions section.

---

> ### Author Rebuttal · Authors · 2024-08-07
>
> We sincerely thank the reviewer for acknowledging the significance of the research problem, the interesting and sound technical design, as well as sufficient experiments. Below we provide point-to-point responses to all the raised questions:
>
> > **Q1: The use of summarization tokens.**
>
> **A1:** Our motivation is to distill the information in a video clip (TN clip tokens) into a compact set of tokens (TP summarization tokens). These TP tokens are intended to serve as a memory that summarizes the clip content, which can be propagated across different clips for long-term memory formulation.
>
> To achieve this, we initialize TP summarization tokens using pooling and feed them into an autoregressive language model. The causal attention layers then aggregate the clip information into the TP tokens to create a compact memory. To ensure these tokens consolidate useful information, we design a prefix task where the language model generates captions using only the TP tokens, without access to the original clip features. This avoids attention between the TT caption tokens and the TN clip tokens, encouraging the TP tokens to encapsulate the clip information.
>
> > **Q2: The necessity of language reasoning for visual-dominant dependencies.**
>
> **A2:** We validate the necessity of language reasoning in Table 2 in rebuttal PDF. We reimplement a conventional video memory strategy MC-ViT [7] for comparison. Our method presents dominant advantages and the reasons are two-fold.
> + MC-ViT consolidates memory according to handcrafted rules like clustering. In contrast, using language model as the streaming encoder allows us to leverage extensive video caption/QA data to guide memory consolidation in an end-to-end manner. This data-driven strategy results in more comprehensive memories that facilitate general video understanding.
> + Since the memory is fed into a subsequent LLM for reasoning, we need to align the memory with the input feature space of the LLM. Compared to the output from ViT backbone, it is easier to align the memory generated by a language model with the feature space of LLM, thus improving performance.
>
> > **Q3: More efficient alternatives in streaming encoder.**
>
> **A3:** We try fewer layers of Phi model as streaming encoder in Table 2 in rebuttal PDF. It indicates too few layers would lead to performance drop due to insufficient memory consolidation. However, even with fewer parameters (0.3B with 4 layers), the language model-based approach still outperforms MC-ViT, which reinforces the necessity of language reasoning for visual feature processing.
>
> > **Q4: The difference in memory propagation with MemDPC.**
>
> **A4:** The major difference lies in the memory content, the memory usage and the technical design.
> + MemDPC formulates a learnable memory bank shared for the entire dataset. The memory serves as the statistics of the whole dataset. In contrast, our memory is a summarization of a specific video.
> + The memory bank in MemDPC is an external knowledge reference that provides multiple future hypotheses for future prediction. Our memory is used to conclude the historical information in a long video to facilitate long-term comprehension.
> + MemDPC only uses memory to augment the future prediction, there is no explicit memory propagation process. Conversely, in our architecture, the memory is propagated across different clips to form a global encapsulation of a long video.
>
> > **Q5: More ablation studies.**
>
> **A5:** Besides Table 8\~9, we have presented ablation studies on summarization tokens, temporal supervision, time prompts, and similarity measurements in Table 10\~13 in the appendix (Page 15\~17).
>
> > **Q6: More baselines like LLaMA-VID.**
>
> **A6:** We add the results of LLaMA-VID with Vicuna-13B on EgoSchema, Next-QA and MovieChat in Table 3 in rebuttal PDF. For MovieChat breakpoint mode, we only input segments of the video up to the breakpoint timestamp to the model. It is easier than our setting which requires adaptive temporal selection. Nevertheless, our model still significantly outperforms LLaVA-VID on all benchmarks.
>
> > **Q7: About progressive training.**
>
> **A7:** Sorry for the confusion. The progressive training part is used to clarify the training process rather than serve as an innovation. We will modify this to make it clearer.
>
> > **Q8: Explanations for significant zero-shot performance on EgoSchema.**
>
> **A8:** The reasons are two-fold.
> + The questions in EgoSchema do not have obvious ego-centric domain-specific characteristics, but rather require long-term temporal memory and perception, which is a major strength of our model.
> + Compared to MC-ViT, our architecture with LLM has stronger generalization ability. It can effectively generalize to different video scenarios even without ego-centric training pairs. This is also validated by the superior zero-shot performance of LLaMA-VID 13B compared to prior methods like InternVideo and FrozenBiLM (35.5 vs 32.1, 35.5 vs 26.9).
>
> > **Q9: About grounding in Figure 5.**
>
> **A9:** For grounding, we calculate the similarity between the question and the time indicators of each clip to select the related segments. Below are some explanations.
> + The similarity-based strategy relies on some semantic relations between questions and clips to do temporal selection. It can correlate the key components in questions like Tower with the clips containing corresponding elements.
> + The grounding intervals are determined by the temporal length of the sampled video clips.
> + We only feed the encoded memories of the selected clips to LLM, so the model fully relies on the grounding results to produce responses.
> + For quantitative evaluation, Next-GQA requires accurate grounding to produce correct answers. The results in Table 5 in the manuscript validate our method reaches promising grounded question-answering ability, even exceeding the methods with specialized grounding modules. There currently lack hour-long grounding QA benchmarks for further evaluation.

---

> > ### Comment · Reviewer_QjSL · 2024-08-12
> > **Post-response**
> >
> > I would like to thank the authors for their hard work and appreciate the honest acknowledgment and significant updates during the rebuttal phase. which have mainly addressed my concerns.
> >
> > Additionally, I suggest the authors to include more discussion on recent works related to long video streaming optimization in their revision, such as [1], which maintaining long context with real-time efficiency.
> >
> > Moreover, recent benchmarks like [2] might offer a better evaluation choice for long videos.
> >
> > [1] Chen, Joya, et al. "VideoLLM-online: Online Video Large Language Model for Streaming Video." Proceedings of the IEEE/CVF Conference on Computer Vision and Pattern Recognition. 2024.
> > [2] Fu, Chaoyou, et al. "Video-MME: The First-Ever Comprehensive Evaluation Benchmark of Multi-modal LLMs in Video Analysis." arXiv preprint arXiv:2405.21075 (2024).
> >
> > Considering these points, I would like to raise my rating to borderline accept.

---

> > > ### Author Response · Authors · 2024-08-12
> > >
> > > Dear Reviewer QjSL:
> > >
> > > We are delighted to hear that your concerns are addressed! Many thanks again for your very constructive comments, which have helped us improve the quality of the paper significantly. We will include detailed discussions with recent works like [1] and incorporate the results on more recent benchmarks like [2].
> > >
> > > Best,
> > >
> > > Paper 6273 Authors.

---

### Official Review · Reviewer_FUM4 · 2024-07-16

**Soundness:** 2
**Presentation:** 3
**Contribution:** 2
**Rating:** 6
**Confidence:** 3

**Summary:**

In this paper, a vision-language model, named VideoStreaming, is proposed.
It adopts the memory based architecture to understand long video. Specifically, it segments a long video into multiple short clips and encodes each clip sequentially. During this encoding process, it exploits the memory feature to propagate the information from previous clips. Also, it develops the memory selection strategy to find out question-related memory among all encoded memories.
Extensive experimental results on various VQA datasets show that the proposed algorithm achieves better performances than exisiting methods.

**Strengths:**

- The paper is well-written and easy to understand
- The motivation is reasonable. Also, the proposed algorithm technically sounds.
- The proposed algorithm achieves

**Weaknesses:**

I could not find the critical flaws in this work.
Even though the motivation is not very new in long video understanding, the proposed memory-based framework is reasonable and well-designed.

- It would be better to provide the comparsion on IntentQA since it is also widely used for LVQA.
- In L201, the authors said 'we develop a variant of Gumbel-Softmax, denoted as Gumbel-TopK(),...' However, there is no detailed description on Gumbel-TopK in the paper and Gumbel-TopK trick is already widely used. Thus, I would recommend using other expression, instead of 'develop.' Otherwise, it would be better to explain Gumbel Top-k in detail.

**Questions:**

In overall, I think that the motivation of the paper is reasonable. Also, the proposed memory propagating encoding framework and memory selection method have some contribution to the field. Therefore, I'm positive about this work. Please find my concerns in the weakness section.

**Limitations:**

Yes, L574-581

---

> ### Author Rebuttal · Authors · 2024-08-07
>
> We sincerely thank the reviewer for recognizing our motivation, technical algorithm, and writing. Below we would like to provide point-to-point responses to all the raised questions:
>
> > **Q1: The comparison on IntentQA dataset.**
>
> **A1:** We compare our model with recent advanced methods in the Table below. We report the zero-shot performance on IntentQA test set. Our method presents a dominant advantage in temporal understanding. And the overall performance is significantly superior to recent works.
>
> | Method | Params | Why | How | Before/After | All |
> | :-: | :-: | :-: | :-: | :-: | :-: |
> | LLaMA-VID | 13B | 43.8 | 40.1 | 36,3 | 41.4 |
> | LLoVi   | 7B  |   57.9  | 55.4  | 42.3 | 53.6 |
> | LangRepo   | 8x7B   | 62.8 | 62.4 | 47.8 | 59.1 |
> | Ours | 7B+1.3B | 65.6 | 66.2 | 59.0 | 64.1 |
>
> > **Q2: The details on Gumbel-TopK.**
>
> **A2:** Sorry for the confusion and missing the detailed explanation for Gumbel-TopK. Similar to the Gumbel-Softmax technique, Gumbel-TopK is used to allow gradient backpropagation when selecting the Top-K indices. The major difference is that Gumbel-TopK can take multiple indices (i.e., the Top-K largest activations) for each sample, while Gumbel-Softmax only takes the index corresponding to the single largest activation for each sample. This design allows us to select multiple video segments that are relevant to the questions (i.e., the Top-K most related ones) to do comprehensive reasoning. This is not a technical contribution, and we will modify 'develop' into 'adopt' for correction.

---

> ### Author Response · Authors · 2024-08-12
> **Additional Response to Reviewer FUM4**
>
> Dear Reviewer FUM4,
>
> We sincerely thank you again for your great efforts in reviewing this paper, especially for the precious advice that has helped us improve the quality of this paper significantly!
>
> In response to your suggestions, we have supplemented the experiments and explanations in the rebuttal. If there are any further clarifications or additional experiments you would like us to address, please do not hesitate to let us know.
>
> Best,
>
> Paper 6273 Authors.

---

> > ### Comment · Reviewer_FUM4 · 2024-08-14
> >
> > I would like thank the authors for their effort for rebuttal. I have read the response from authors as well as the reviews from other reviewers. The rebuttal has resolved most of my concerns. Therefore, I decided to keep my original rating.

---

> > > ### Author Response · Authors · 2024-08-14
> > >
> > > Dear Reviewer FUM4:
> > >
> > > We are delighted to hear that your concerns are addressed! Many thanks again for your very constructive comments, which have helped us improve the quality of the paper significantly. We will include the experiments and clarifications in the revised version.
> > >
> > > Best,
> > >
> > > Paper 6273 Authors.

---

### Author Rebuttal · Authors · 2024-08-07

Dear reviewers,

We sincerely appreciate the constructive feedbacks provided by all the reviewers. The reviewers acknowledged some aspects of our work, including the motivation and novelty (Reviewer FUM4, sqPz), the significance of the research problem (Reviewer QjSL), the technical design (Reviewer FUM4, QjSL, sqPz), the thorough experiments (Reviewer QjSL), the efficiency in handling long videos (Reviewer 9Ye5), as well as the paper writing (Reviewer FUM4, sqPz).

Below we summarize some major points we have addressed in the rebuttal. More detailed responses are provided individually for each reviewer.

**More experimental comparisons:** We complement the experiments on IntentQA dataset (Reviewer FUM4), the comparison with LLaMA-VID on EgoSchema, Next-QA and MovieChat (Reviewer QjSL) and the comparison with VideoChat2 (Reviewer sqPz).

**More ablation studies and explanations:** We add the ablation studies on the instantiation of streaming encoder (Reviewer QjSL, sqPz), more ablation studies on hour-long MovieNet-QA dataset (Reviewer sqPz), and the explanations of different settings (Reviewer sqPz).

**More insights to the streaming encoder:** We provide further explanations on the design of summarization tokens (Reviewer QjSL), the necessity of language reasoning in memory formulation (Reviewer QjSL), as well as the influence of using different number of layers and different models as the streaming encoder (Reviewer QjSL, sqPz).

**Further explanations on the example of hour-long video QA:** We further explain the temporal grounding process (Reviewer QjSL, sqPz), and reformulate the question to avoid information leakage that may make LLM recall the text pretraining data for comparison (Reviewer sqPz).

**More discussions with realted works:** We clarify the differences between our propagated memory and the memory bank design in MemDPC (Reviewer QjSL), explain the significant zero-shot performance on EgoSchema compared to MC-ViT(Reviewer QjSL), and add discussion with LGDN on the clip selection process (Reviewer 9Ye5).

**Clarifications on some confusing points:** We provide clarifications on some confusing points, including the description of Gumbel-TopK technique (Reviewer FUM4), the progressive training (Reviewer QjSL), the efficiency of handling multiple queries per video (Reviewer 9Ye5), and the description in L127-L129 (Reviewer sqPz).

Due to space constraints, we have placed some of the added Figures and Tables in the uploaded author rebuttal PDF. In the detailed point-to-point responses below, we have specified the location of referred Figures and Tables. Please don’t hesitate to let us know if there are any additional clarifications or experiments that we can offer!

Yours Sincerely,

Authors

---

### Decision · Program_Chairs · 2024-09-25

**Decision:**

Accept (poster)

**Comment:**

The authors proposed a model for streaming video understanding. Specifically, the authors break up a long video into multiple short clips, and encode each clip sequentially one-after-the-other, using a memory module to propagate information from one clip to the next. The authors also develop a memory selection method to extract the features from the memory which are most relevant to the caption. The authors show strong results on a number of video question-answering datasets.

During the rebuttal, the authors addressed the concerns from the reviewers, which led to most of them increasing their score. Ensure that all of these revisions are in the final camera-ready version of the paper. In the camera-ready version, please also discuss the relation to the memory model of [this recent published work](https://openaccess.thecvf.com/content/CVPR2024/papers/Zhou_Streaming_Dense_Video_Captioning_CVPR_2024_paper.pdf) on streaming video understanding.